# Serum Klotho in Living Kidney Donors and Kidney Transplant Recipients: A Meta-Analysis

**DOI:** 10.3390/jcm9061834

**Published:** 2020-06-12

**Authors:** Charat Thongprayoon, Javier A. Neyra, Panupong Hansrivijit, Juan Medaura, Napat Leeaphorn, Paul W. Davis, Wisit Kaewput, Tarun Bathini, Sohail Abdul Salim, Api Chewcharat, Narothama Reddy Aeddula, Saraschandra Vallabhajosyula, Michael A. Mao, Wisit Cheungpasitporn

**Affiliations:** 1Division of Nephrology and Hypertension, Mayo Clinic, Rochester, MN 55905, USA; chewcharat.api@mayo.edu; 2Division of Nephrology, Bone and Mineral Metabolism, Department of Internal Medicine, University of Kentucky, Lexington, KY 40506, USA; javier.neyra@uky.edu; 3Charles and Jane Pak Center for Mineral Metabolism and Clinical Research, Dallas, TX 75390, USA; 4Division of Nephrology, Department of Internal Medicine, University of Texas Southwestern Medical Center, Dallas, TX 75390, USA; 5Department of Internal Medicine, University of Pittsburgh Medical Center Pinnacle, Harrisburg, PA 17105, USA; hansrivijitp@upmc.edu; 6Division of Nephrology, Department of Medicine, University of Mississippi Medical Center, Jackson, MS 39216, USA; jmedaura@umc.edu (J.M.); pwdavis@umc.edu (P.W.D.); sohail3553@gmail.com (S.A.S.); 7Renal Transplant Program, University of Missouri-Kansas City School of Medicine/Saint Luke’s Health System, Kansas City, MO 64110, USA; napat.leeaphorn@gmail.com; 8Department of Military and Community Medicine, Phramongkutklao College of Medicine, Bangkok 10400, Thailand; wisitnephro@gmail.com; 9Department of Internal Medicine, University of Arizona, Tucson, AZ 85721, USA; tarunjacobb@gmail.com; 10Division of Nephrology, Department of Medicine, Deaconess Health System, Evansville, IN 47710, USA; dr.anreddy@gmail.com; 11Department of Cardiovascular Medicine, Mayo Clinic, Rochester, MN 55905, USA; Vallabhajosyula.Saraschandra@mayo.edu; 12Division of Nephrology and Hypertension, Mayo Clinic, Jacksonville, FL 32224, USA; mao.michael@mayo.edu

**Keywords:** klotho, α-Klotho, FGF-23, kidney transplantation, kidney donor, renal transplantation, transplantation, Nephrology, CKD-MBD, CKD-Mineral and Bone Disorder

## Abstract

α-Klotho is a known anti-aging protein that exerts diverse physiological effects, including phosphate homeostasis. Klotho expression occurs predominantly in the kidney and is significantly decreased in patients with chronic kidney disease. However, changes in serum klotho levels and impacts of klotho on outcomes among kidney transplant (KTx) recipients and kidney donors remain unclear. A literature search was conducted using MEDLINE, EMBASE, and Cochrane Database from inception through October 2019 to identify studies evaluating serum klotho levels and impacts of klotho on outcomes among KTx recipients and kidney donors. Study results were pooled and analyzed utilizing a random-effects model. Ten cohort studies with a total of 431 KTx recipients and 5 cohort studies with a total of 108 living kidney donors and were identified. After KTx, recipients had a significant increase in serum klotho levels (at 4 to 13 months post-KTx) with a mean difference (MD) of 243.11 pg/mL (three studies; 95% CI 67.41 to 418.81 pg/mL). Although KTx recipients had a lower serum klotho level with a MD of = −234.50 pg/mL (five studies; 95% CI −444.84 to −24.16 pg/mL) compared to healthy unmatched volunteers, one study demonstrated comparable klotho levels between KTx recipients and eGFR-matched controls. Among kidney donors, there was a significant decrease in serum klotho levels post-nephrectomy (day 3 to day 5) with a mean difference (MD) of −232.24 pg/mL (three studies; 95% CI –299.41 to −165.07 pg/mL). At one year following kidney donation, serum klotho levels remained lower than baseline before nephrectomy with a MD of = −110.80 pg/mL (two studies; 95% CI 166.35 to 55.24 pg/mL). Compared to healthy volunteers, living kidney donors had lower serum klotho levels with a MD of = −92.41 pg/mL (two studies; 95% CI −180.53 to −4.29 pg/mL). There is a significant reduction in serum klotho levels after living kidney donation and an increase in serum klotho levels after KTx. Future prospective studies are needed to assess the impact of changes in klotho on clinical outcomes in KTx recipients and living kidney donors.

## 1. Introduction

α-Klotho (klotho) is a membrane protein that is highly expressed in the kidney, especially in the distal tubular epithelial cells [1,2,3,4,5,6,7,8,9,10]. Membrane-bound klotho regulates phosphate homeostasis by acting as a co-factor of fibroblast growth factor 23 (FGF23) [11,12,13,14]. FGF23-Klotho signaling promotes urinary phosphate excretion and suppresses the expression of renal 1α-hydroxylase, resulting in reduced vitamin D-dependent intestinal absorption of calcium and phosphate [11,15]. Altogether, FGF23-Klotho signaling regulates phosphate metabolism and prevents phosphate retention [16,17,18,19,20]. Soluble klotho can be detected in the circulation in two forms: (1) cleaved klotho, which is derived from cleavage of the extracellular domain of membrane klotho, and potentially (2) secreted klotho, which is derived from an alternatively spliced klotho mRNA transcript [21,22].

Soluble klotho displays diverse physiological effects and hormonal functions, including the reduction of oxidative stress and the inhibition of intracellular insulin and insulin-like growth factor 1 (IGF-1) signaling [15,23,24,25,26,27,28]. Klotho protects the kidney by suppression of apoptosis [29,30] and cell senescence [31,32], suppression of fibrosis [33,34,35,36,37], and upregulation of autophagy [3,38] in renal tubular cells. Klotho-deficient mice develop premature aging, hyperphosphatemia, vascular calcification and endothelial dysfunction, and have shorter lifespans, while klotho overexpressing mice have 20–30% longer lifespans than wild type mice [2,24,39]. Since klotho expression is the most abundant in the kidney [40], patients with kidney diseases, including acute kidney injury (AKI) and chronic kidney disease (CKD), are found to have a significant reduction in klotho expression and soluble levels [41,42,43,44,45,46,47,48,49,50,51]. Studies have demonstrated that serum klotho declines in progressive human CKD with the lowest serum klotho levels among patients with end-stage kidney disease (ESKD) on dialysis [41,48]. Low serum klotho is associated with increased mortality and cardiovascular events among patients with ESKD [52].

When compared to treatment with chronic dialysis, kidney transplantation (KTx) is the best therapeutic option for patients with ESKD and is associated with increased survival and better quality of life [53,54,55,56]. In addition, living donor KTx provides greater allograft longevity than those transplanted from a deceased donor [57]. However, changes in serum klotho levels and the impact of klotho on outcomes among KTx recipients and kidney donors remain unclear [58,59,60,61,62,63,64,65,66,67,68,69,70,71,72,73,74,75]. Thus, we conducted this systematic review and meta-analysis to assess serum klotho levels and the impact of klotho on outcomes among KTx recipients and kidney donors

## 2. Methods

### 2.1. Search Strategy and Literature Review

A systematic literature search of MEDLINE (1946 to October 2019), EMBASE (1988 to October 2019), and the Cochrane Database of Systematic Reviews (database inception to October 2019) was conducted (1) to assess studies evaluating serum klotho levels and effects of klotho on outcomes among KTx recipients and kidney donors. The systematic literature review was undertaken independently by two investigators (C.T. and W.C.) using a search strategy that combined the terms of (“klotho” OR “klotho protein” OR “klotho gene”) AND (“kidney transplantation” OR “renal transplantation” OR “kidney donor”) which is provided in online Appendix A. No language limitation was applied. A manual search for conceivably relevant studies using references of the included articles was also performed. This study was conducted by the PRISMA (Preferred Reporting Items for Systematic Reviews and Meta-Analysis) statement [76]. The data for this meta-analysis are publicly available through the Open Science Framework (URL: https://osf.io/kx9we/).

### 2.2. Selection Criteria

Eligible studies must have been (1) clinical trials or observational studies (cohort, case-control, or cross-sectional studies) that evaluated serum klotho levels and effects of klotho on outcomes among KTx recipients or kidney donors, and (2) studies that presented data to calculate mean differences (MDs) with 95% confidence intervals (CIs) that evaluated changes in serum klotho before and after KTx/kidney donation or compared serum klotho between KTx patients/donors and a control group composed of non-KTx or non-donor controls. Retrieved articles were individually reviewed for eligibility by the two investigators (C.T. and W.C.). Discrepancies were addressed and solved by joint consensus. Inclusion was not limited by the size of the study.

### 2.3. Data Abstraction

A structured data collecting form was used to obtain the following information from each study including the title, name of the first author, publication year, year of the study, country where the study was conducted, demographic data of kidney transplant recipients and donors, methods used to measure serum klotho, serum klotho levels, estimated glomerular filtration rate (eGFR), control group, and adjusted effect estimates with 95% CI and covariates that were adjusted for in the multivariable analysis. This data extraction process was independently performed by two investigators (C.T. and W.C.).

### 2.4. Statistical Analysis

Analyses were performed utilizing the Comprehensive Meta-Analysis 3.3 software (version 3; Biostat Inc, Englewood, NJ, USA). Adjusted point estimates from each study were consolidated by the generic inverse variance approach of DerSimonian and Laird, which designated the weight of each study based on its variance [77]. The summary statistics for each outcome were the mean change from baseline and standard deviation (SD) of the mean change. The mean change in each group was obtained by subtracting the final mean from the baseline mean. The MDs were preferred since all studies use the same continuous outcome and unit of measure (pg/mL) of serum klotho and FGF-23 levels. The SD of mean change was computed, assuming a conservative correlation coefficient of 0.5 [78]. Effects sizes of 0.2 were interpreted as small, those of 0.5 as moderate, and of 0.8 as large [79]. Given the possibility of between-study variance, we used a random-effect model rather than a fixed-effect model. Cochran’s Q test and I^2^ statistics were applied to determine between-study heterogeneity. A value of I^2^ of 0% to 25% represents insignificant heterogeneity, 26% to 50% low heterogeneity, 51% to 75% moderate heterogeneity and 76–100% high heterogeneity [80]. The presence of publication bias was assessed by the Egger test [81].

## 3. Results

A total of 132 potentially eligible articles were identified using our search strategy. After the exclusion of 93 articles based on title and abstract for clearly not fulfilling inclusion criteria on the basis of the type of article, study design, population or outcome of interest, or due to being duplicates, 39 articles were left for full-length review. Eighteen of these were excluded from the full-length review as they did not report the outcome of interest, while six articles were excluded because they were not observational studies. Thus, 15 studies (10 cohort studies [58,59,60,61,62,63,64,65,66,67] with a total of 431 KTx recipients and 5 cohort studies [68,69,70,71,72] with a total of 108 living kidney donors) were included. The literature retrieval, review, and selection process are demonstrated in Figure 1.

### 3.1. Serum Klotho after Kidney Transplantation

The characteristics of the included studies assessing serum klotho after kidney transplantation are presented in Table 1 and Table 2. After KTx, there was a significant increase in serum klotho levels in recipients (at 4 to 13 months post-KTx) in reference to baseline levels before KTx with a mean difference (MD) of 243.11 pg/mL (three studies; 95% CI 67.41 to 418.81 pg/mL, I^2^ = 93%), Figure 2A. There were significant reductions in serum PTH and phosphate levels with MDs of −134.65 pg/mL (95% CI −176.09 to −93.21 pg/mL, I^2^ = 0%) and −2.81 mg/dL (95% CI −3.46 to −2.16 mg/dL, I^2^ = 97%), respectively. There was no significant change in serum calcium levels with a MD of 0.37 mg/dL (95% CI, −0.05 to 0.79 mg/dL, I^2^ = 83%). Although KTx recipients had lower serum klotho levels with a MD of = −234.50 pg/mL (five studies; 95% CI −444.84 to −24.16 pg/mL, I^2^ = 93%, Figure 2B) compared to healthy unmatched volunteers, one study demonstrated comparable klotho level between KTx recipients and eGFR-matched controls [66]. Two studies demonstrated high serum klotho levels in deceased donors as a prognostic marker for good allograft function within one year after KTx (*p* < 0.05) [59,60].

### 3.2. Serum Klotho after Living Kidney Donation

The characteristics of the included studies assessing serum klotho after kidney transplantation are presented in Table 3 and Table 4. A total of 108 living kidney donors were identified from five cohort studies. After kidney donation, there was a significant decrease in serum klotho levels post-nephrectomy (day 3 to day 5) with a mean difference (MD) of −232.24 pg/mL (three studies; 95% CI −299.41 to −165.07 pg/mL, I^2^ = 0), Figure 3A. At one year following the kidney donation, serum klotho levels remained lower than baseline before nephrectomy with a MD of = −110.80 pg/mL (two studies; 95% CI −166.35 to −55.24 pg/mL, I^2^ = 5), Figure 3B.

There was no significant change in serum FGF-23 at one year post-donation with a MD of = 8.19 pg/mL (two studies; 95% CI −14.24 to 30.62 pg/mL, I^2^ = 85%), Figure 4A. Compared to unmatched healthy volunteers, living kidney donors had lower serum klotho levels with a MD of = −92.41 pg/mL (two studies; 95% CI −180.53 to −4.29 pg/mL, I^2^ = 44%), Figure 4B.

### 3.3. Evaluation for Publication Bias

A funnel plot was not drawn because of the limited number of studies in each analysis. Generally, tests for funnel plot asymmetry should be used only when there are at least ten study groups, because the power of the test is too low to distinguish chance from real asymmetry [82]. Egger’s regression test demonstrated no significant publication bias in all analyses (*p* > 0.05).

## 4. Discussion

In this meta-analysis, we demonstrated that serum klotho levels were significantly increased after successful KTx. While KTx recipients had lower serum klotho levels compared to unmatched healthy volunteers, serum klotho levels in kidney transplant recipients were comparable to those in eGFR-matched controls. Among kidney donors, we found a significant decrease in serum klotho levels post-nephrectomy at day 3 to day 5, which remained lower than baseline before nephrectomy at one year following kidney donation. Compared to healthy volunteers, living kidney donors had lower serum klotho levels.

The findings from our meta-analysis support that klotho is primarily synthesized in the kidneys [40], and transplanting a new kidney into ESKD patients would result in an increase in renal klotho and serum klotho levels post-KTx. In addition to the oligo-anuric state, patients with advanced CKD/ESKD have a significant reduction in klotho and progressively lose the ability to prevent phosphate retention, resulting in hyperphosphatemia, vascular calcification, and cardiovascular disease [83,84]. After successful KTx, in addition to improvement in eGFR, there is also a significant increase in klotho, altogether leading to an improvement in phosphate homeostasis. Recent studies have demonstrated that post-transplant hypophosphatemia after KTx is associated with good kidney allograft function [85,86]. Although the actual underlying mechanisms remain unclear, this is likely because excellent quality transplanted kidneys have higher eGFR and klotho expression, resulting in a reduction in phosphate levels post-KTx.

We identified two cohorts of KTx patients who received their kidneys from deceased donors; higher serum klotho levels in these donors were prognostic for good allograft function at one year after KTx [59,60]. In the ischemia-reperfusion injury (IRI), which is unavoidable to a certain degree in all KTx surgeries, soluble klotho protects renal tubular cells from oxidative damage by inhibiting the insulin/IGF-1 signaling pathway and by inhibition of TGF-β1 for decreasing renal fibrosis [87,88], and upregulation of autophagy in renal tubular cells [3,89]. In addition, klotho is also involved in the inhibition of Wnt pathway-associated β-catenin activation, thus improving renal fibrosis [87]. Compared to patients with early graft function, a lower level of klotho is observed in implantation biopsies among patients with delayed graft function (DGF) [90]. Although data on the effects of klotho on long-term allograft outcomes are limited, it is well known that poor allograft function at one year after KTx and DGF is associated with renal allograft loss [91,92]. Following successful KTx, patients regain functions of klotho via FGF23-Klotho signaling, and with the previously accumulated FGF23, residual hyperparathyroidism, and the use of calcineurin inhibitors (especially cyclosporine) [93,94,95], post-KTx hypophosphatemia can commonly occur up to 86% [85,96,97]. Post-KTx hypophosphatemia is known to be associated with lower risks of death-censored graft failure and cardiovascular mortality [85]. The association between post-KTx hypophosphatemia and reduced cardiovascular mortality among KTx recipients could be related to the reduction of calcium phosphate product, an important factor associated with vascular calcification and cardiovascular events [98,99]. Our study demonstrated that successful KTx can result in a significant increase in serum klotho levels among KTx recipients [85]. In addition, previous literature has demonstrated trending towards normal FGF-23 levels after successful KTx [42,100]. Thus, regaining function in FGF23-Klotho signaling after KTx helps promote urinary phosphate excretion and reduced vitamin D-dependent intestinal absorption of calcium and phosphate [11,15], which might explain the association between post-KTx hypophosphatemia and reduced cardiovascular mortality. Future studies are needed to assess the impact of klotho levels on long-term cardiovascular health in KTx recipients, allograft, and patient survival.

Living donors supply approximately 40% of kidney allografts in the United States [101]. Overall, living kidney donation is considered safe and does not appear to increase long-term mortality compared with controls [102,103,104,105,106,107]. A recent systematic review of 52 studies comprising 118,426 living kidney donors reassured the safety of living kidney donations with the finding of no difference in all-cause mortality among donors and controls [108]. In addition, a large retrospective population-based matched cohort study of 2028 kidney donors in comparison with 20,280 matched non-donor controls (followed for a median of 6.5 years) demonstrated no difference in the rate of cardiovascular events between the two groups [109]. Although the findings of our study showed a significant reduction in serum klotho at post-operative day 3 to 5 and at one year following kidney donation, the degree of klotho reduction seemed to be attenuated at one year post-donation compared to the early post-operative period. In addition, we found no significant change in serum FGF-23 at one year post-donation. It is possible that after living kidney donation serum klotho is not severely reduced enough to stimulate the rise in serum FGF-23, which occurred in patients with advanced CKD [41,110]. Elevated FGF-23 levels have been shown to be associated with increased mortality and cardiovascular events [111,112,113]. Thus, no significant increase in FGF-23 levels after living kidney donation is consistent with the findings of no difference in all-cause mortality among donors and controls in previous literature [108,109].

Despite these published reassuring findings of donor safety [108,109], a recent small multicenter study of living kidney donors and healthy controls (n = 124) demonstrated an association between living kidney donation and a significant increase in left ventricular mass and reduced aortic distensibility [114]. In addition to functions of klotho via FGF23-Klotho signaling, soluble klotho also has FGF23-independent effects, including endothelial protection from senescence, anti-fibrotic properties, cardioprotection, and prevention of vascular calcifications [84,115,116]. Klotho-deficient CKD mice have significant left ventricular hypertrophy (LVH) and cardiac fibrosis compared with wild-type mice [117]. Soluble klotho also provides cardioprotection against stress-induced exaggerated cardiac remodeling through downregulation of transient receptor potential cation channel 6 (TRPC6) [118]. Although an increased LVH and reduced aortic distensibility in living kidney donors could be related to an increased risk of hypertension post living kidney donation [102,103], future studies are required to assess whether a reduction in serum klotho levels after living kidney donation may play a role in the higher risk of LVH, and reduced aortic distensibility observed among living kidney donors.

Our meta-analysis is subject to certain limitations. First, although there were comparative groups, all studies are observational, making them susceptible to selection bias. Second, many variables may influence klotho levels in the post-transplant period that may contribute to the heterogeneity between the included studies evaluating changes in serum klotho levels among KTx recipients. Data on medications that may affect endogenous klotho expression in the kidney and soluble levels such as angiotensin II inhibitors and hydroxymethylglutaryl-CoA (HMG-CoA) reductase inhibitors [17,21,119,120] as well as data on immunosuppression were limited in included studies. Lifestyle, diet, psychological stress, and activities such as exercises may also affect serum klotho levels [121,122,123,124]. Thus, future prospective studies are needed to assess the impact of changes in klotho on clinical outcomes in KTx recipients and living kidney donors. Third, the follow-up duration of included studies was limited to only one year, and future studies are required to evaluate the impacts of serum klotho levels on long-term clinical outcomes. Fourth, serum klotho is also affected by the aging process and declines with older age [125]. However, we demonstrated an increase in serum klotho levels after KTx at one year and a decrease in klotho levels at immediate postoperative (which is less likely to be affected by the aging process). Lastly, all included studies measured serum klotho levels by ELISA. Recently, immunoprecipitation-immunoblot (IP-IB) assay is shown to be superior to the ELISA and highly correlated with eGFR [126]. However, this technique requires the labor-intensive nature of the IP-IB assay, and further research is needed to evaluate the use of the IP-IB assay in KTx patients.

In conclusion, compared to patients’ baseline, serum klotho levels increase early after successful KTx and decrease after living kidney donation, respectively. Future studies are required to assess the impact of serum klotho levels on risk-stratification and patient-centered outcomes in both living donors and KTx recipients.

## Figures and Tables

**Figure 1 jcm-09-01834-f001:**
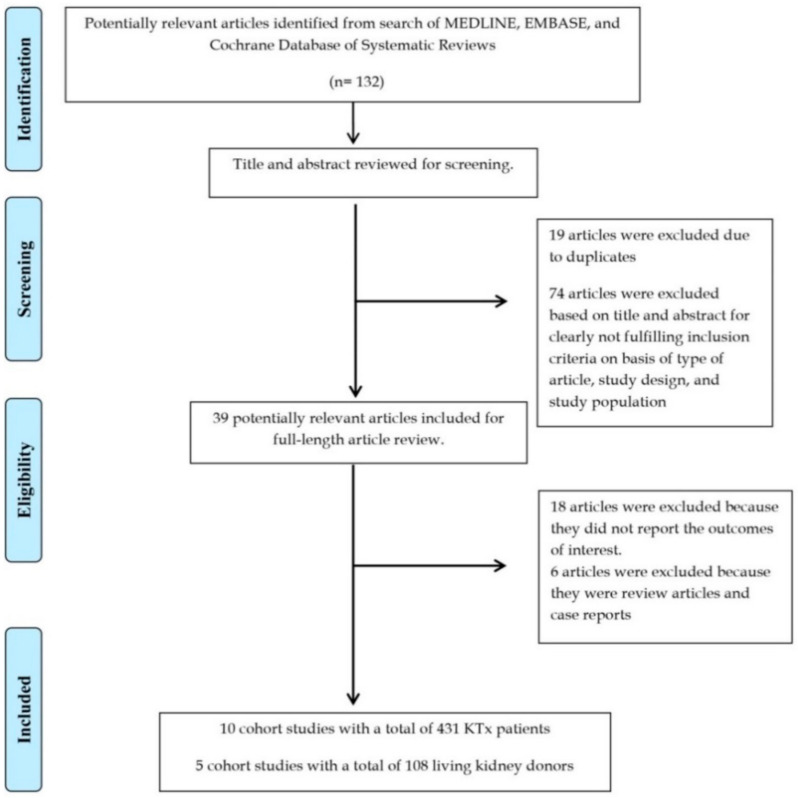
Outline of our search methodology. Abbreviation: KTx, kidney transplant.

**Figure 2 jcm-09-01834-f002:**
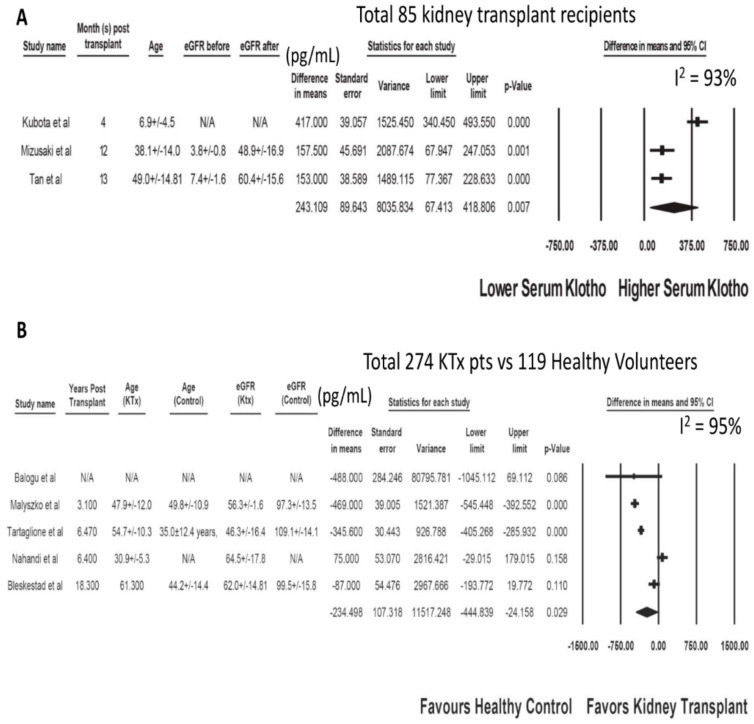
(**A**) Change in Serum Klotho in KTx Recipients after Kidney Transplant. (**B**) Serum Klotho in KTx Recipients Compared to Unmatched Healthy Volunteers.

**Figure 3 jcm-09-01834-f003:**
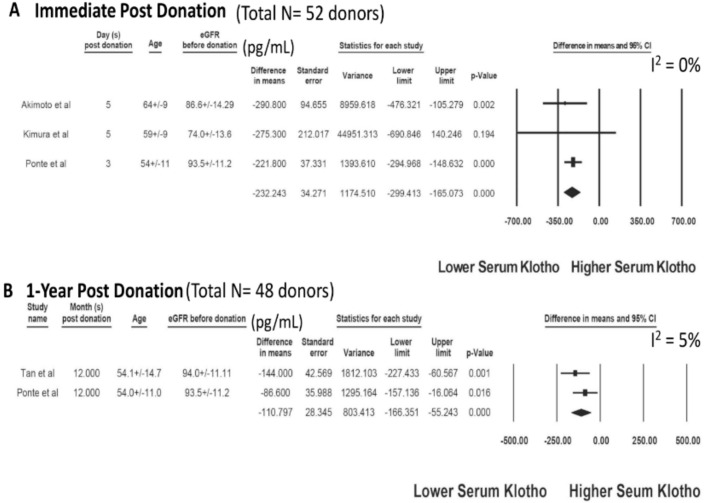
Changes in serum klotho after living kidney donation: (**A**) immediate post-donation and (**B**) one year post-donation.

**Figure 4 jcm-09-01834-f004:**
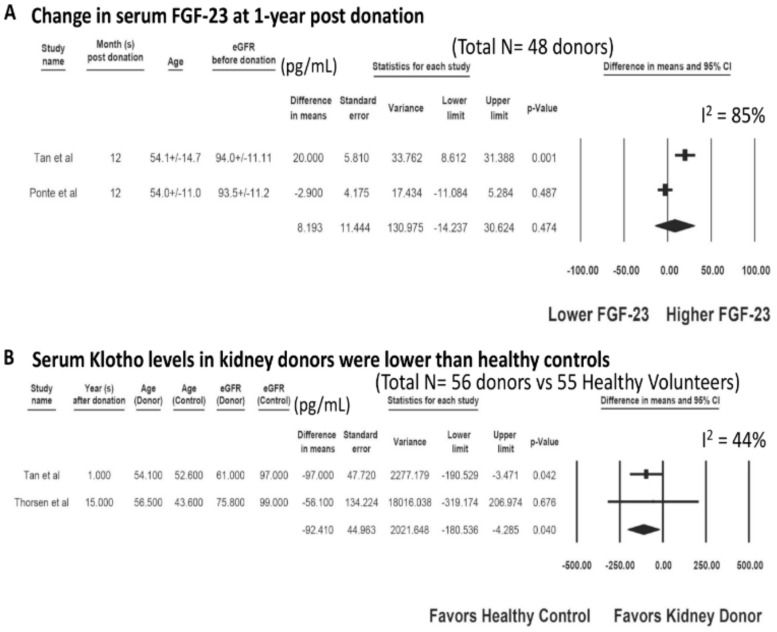
(**A**) Changes in Serum FGF-23 at one year post-donation and (**B**) Serum klotho levels in kidney donors compared to unmatched healthy controls.

**Table 1 jcm-09-01834-t001:** Characteristics of the included studies assessing serum klotho after kidney transplantation.

Study	Year	Country	N-KTx	Characteristics-KTx	Klotho before KTx (pg/mL)	Other Markers before KTx	Kloth after KTx (pg/mL)	Other Markers after KTx
Kubota et al.		Japan	20	Age 6.9 ± 4.5 years	988 ± 122	FGF23	At 4 months	N/A
				Male 12 (60%)		5343 ± 1350 pg/mL	1405 ± 125	
Tan et al.	2017	Australia	29	Age 49 (35–55) years	307 (279–460)	iFGF23	At 52 weeks	At 52 weeks
								iFGF23
						2060 (825–5075) pg/mL		64 (34–88) pg/mL
						eGFR	460 (311–525)	eGFR
				Male 17 (59%)		7.4 (6.5–8.7) mL/min/1.73 m^2^		60.4 (50.5–71.6) mL/min/1.73 m^2^
Mizusaki et al.	2019	Japan	36	Age 38.1 ± 14 years	211.8	eGFR	At 1 year	At 1 year
								eGFR
				Male 15 (42%)		3.8 ± 0.8 mL/min/1.73 m^2^	369.3	49 ± 17 mL/min/1.73 m^2^

Abbreviations: eGFR, estimated glomerular filtration rate; iFGF23, intact fibroblast growth factor-23; KTx, kidney transplant; N/A, not available.

**Table 2 jcm-09-01834-t002:** Characteristics of the included studies comparing serum klotho between KTx recipients and healthy volunteers.

Study	Year	Country	N-KTx	Characteristics-KTx	Klotho-KTx (pg/mL)	Other Markers-KTx	N-Control	Klotho-Control (pg/mL)	Other Markers-Control
Balogu et al.		Turkey	40	N/A	153 ± 170	FGF23	20 healthy subjects	641 ± 1797	FGF23
						47.4 ± 61 pg/mL			1.6 ± 1.3 pg/mL
Malyszko et al.	2014	Poland	84	Median time from KTx	228 (161–384)	eGFR	22 healthy subjects	757 (632–839)	eGFR
						56.3 ± 1.6			
				37 (13–72) months		ml/min/1.73 m2			97.3 ± 13.5 mL/min/1.73 m^2^
				Age 47.9 ± 12.0 years		FGF23			FGF23
				Male 64 (76%)		16.7 (13.8–21.2) pg/mL			11.7 (10.8–17.2) pg/mL
Bleskestad et al.	2015	Norway	40	Median time from KTx	605 (506–784)	eGFR	39 GFR-matched controls	GFR-matched controls	GFR-matched control
									eGFR
									62 (57–73) mL/min/1.73 m^2^
									iFGF23
									63 (52–87) pg/mL
									Healthy volunteer
									eGFR
									99.5 (89.5–110.8)
				18.3 (IQR 12.2–26.2) years		62 (52–72) mL/min/1.73 m^2^		660 (536–847)	mL/min/1.73 m^2^
				Age 61.3 ± 11.8 years		iFGF23	20 healthy subjects	Healthy volunteers	iFGF23
				Male 29 (73%)		75 (53–108) pg/mL		692 (618–866)	51 (36–68) pg/mL
Tartaglione et al.	2017	Italy	80	Time for KTx	449 (388–534)	eGFR	30 healthy subjects	795 (619–901)	eGFR
				77.6 (37.6–119.5) months		46.3 (36.2–58.3) mL/min/1.73 m2			109.1 ± 14.1 mL/min/1.73 m^2^
				Age 54.7 ± 10.3 years		FGF23			FGF23
				Male 49 (61%)		41 (25–59) pg/mL			34 (28–441) pg/mL
Nahandi et al.	2017	Iran	30	Time from KTx	276 ± 241	eGFR	27 healthy subjects	N/A	N/A
				6.42 ± 2.44 years		64.53 ± 17.83 mL/min/1.73 m^2^			
				Age 30.9 ± 5.3 years					

Abbreviations: eGFR, estimated glomerular filtration rate; FGF23, fibroblast growth factor-23; KTx, kidney transplant; N/A, not available.

**Table 3 jcm-09-01834-t003:** Characteristics of the included studies evaluating serum klotho after living kidney donation.

Study	Year	Country	N-Donor	Characteristics-Donor	Klotho before Donor Nephrectomy (pg/mL)	Other Markers before Donor Nephrectomy	Klotho after Donor Nephrectomy (pg/mL)	Other Markers after Donor Nephrectomy
Akimoto et al.	2013	Japan	10	Age 64 ± 9 years	910 (755–1132)	eGFR	At day 5	N/A
				Male 4 (40%)		87 (72–92) mL/min/1.73 m^2^	619 (544.6–688.5)	
				Living donor				
Ponte et al.	2014	Switzerland	27	Age 54 ± 11 years	526 (482–615)	eGFR	At day 3	At day 3
								FGF23
								26.9 (22.1–38.0) pg/mL
								At day 360
								eGFR
						95 ± 11 mL/min/1.73 m^2^	304 (266–491)	63 ± 13 mL/min/1.73 m^2^
				Male 15 (57%)		FGF23	At day 360	FGF23
				Living donor		48.1 (37.4–60.0) pg/mL	440 (398–613)	45.2 (37.7–56.4) pg/mL
Kimura et al.	2015	Japan	15	Age 59 ± 9 years	1084 (795–1638)	eGFR	At day 5	N/A
				Male 8 (53%)		74 ± 14 mL/min/1.73 m^2^	809 (638–1357)	
				Living donor				
Tan et al.	2017	Australia	21	Age 54.1 ± 14.7	564 (468–663)	eGFR	At 12 months	At 12 months
								eGFR
						94 (82–97) mL/min/1.73 m^2^		61 (49–69) ml/min/1.73 m^2^
				Male 13 (62%)		iFGF23	420 (378–555)	iFGF23
				Living donor		52 ± 15 pg/mL		72 ± 22 pg/mL

Abbreviations: eGFR, estimated glomerular filtration rate; FGF23, fibroblast growth factor-23; N/A, not available.

**Table 4 jcm-09-01834-t004:** Characteristics of the included studies comparing serum klotho between living kidney donors and healthy volunteers.

Study	Year	Country	N-Donor	Characteristics-Donor	Klotho-Donor (pg/mL)	Other Markers-Donor	N-Control	Klotho-Control (pg/mL)	Other Markers-Control
Thorsen et al.	2016	Norway	35	Age 56.5 ± 9.4 years	669 (409–1161)	FGF23	35 healthy subjects	725 (458–1222)	FGF23
				Male 21 (60%)		62.6 (6.6–112) pg/mL			51.8 (25.9–90) pg/mL
				Living donor					
				Time from donor nephrectomy		eGFR			eGFR
				Median 15 years		75.8 ± 12.3 mL/min/1.73 m^2^			99.0 ± 13.1 mL/min/1.73 m^2^
Tan et al.	2017	Australia	21	Age 54.1 ± 14.7 years	420 (378–555)	eGFR	20 healthy subjects	517 (434–667)	eGFR
				Male 13 (62%)		61 (49–69) mL/min/1.73 m^2^			97 (89–102) mL/min/1.73 m^2^
				Living donor		iFGF23			iFGF
				Time from donor nephrectomy 1 year		72 ± 22 pg/mL			52 ± 15 pg/mL

Abbreviations: eGFR, estimated glomerular filtration rate; FGF23, fibroblast growth factor-23.

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
