# Peer review of "Serum Klotho in Living Kidney Donors and Kidney Transplant Recipients: A Meta-Analysis"

_jcm, 2020, doi:10.3390/jcm9061834_

Round 1
Reviewer 1 Report
The study addresses an interesting question. Considering the available studies, the meta-analysis is well-conducted. The discussion of the results is nice, and some limitations are noted by the authors. Finally, the manuscript is well-written. Thus, the paper can be published after some improvements in the discussion.
- The authors rightly wrote that decreased Klotho contributes to hyperphosphatemia in advanced CKD. However, this is not applied to most of the involved in the study patients. Considering that before transplantation, most of these patients were on hemodialysis treatment, the primary reason for hyperphosphatemia should have been their oligo-anuric state.
- As regards the reason post-KTx hypophosphatemia, the study showed that post-KTx Klotho levels are lower than controls, while FGF23 similar to the control. Thus, the given explanation only partially can be the reason for post-KTx hypophosphatemia. And certainly due to ischemia-reperfusion injury, the quality of the grafts is not more excellent than of a native normal kidney. Initially, post-KTx hypophosphatemia can be partly attributed to the previously accumulated FGF23, but there are also other more persistent reasons. Residual hyperparathyroidism, which is not uncommon, is another explanation. Cyclosporine may also induce hypophosphatemia since it inhibits the proximal tubular brush border membrane Na/Pi cotransport activity in rats. A study detected faster recovery of impaired phosphate reabsorption by the proximal tubule in patients treated with tacrolimus compared to cyclosporine. These points were not evaluated and should be discussed reasonably instead of indirectly, through hypophosphatemia, attribute the better outcome to partial Klotho restoration after KTx.
Author Response
Response to Reviewer#1
The study addresses an interesting question. Considering the available studies, the meta-analysis is well-conducted. The discussion of the results is nice, and some limitations are noted by the authors. Finally, the manuscript is well-written. Thus, the paper can be published after some improvements in the discussion..
Response: We thank you for reviewing our manuscript and for your critical evaluation. We hope these revisions will satisfy your standards.
Comment #1
The authors rightly wrote that decreased Klotho contributes to hyperphosphatemia in advanced CKD. However, this is not applied to most of the involved in the study patients. Considering that before transplantation, most of these patients were on hemodialysis treatment, the primary reason for hyperphosphatemia should have been their oligo-anuric state
Response: We appreciate the reviewer’s important point. We agree with the reviewer, and thus we have added the point of oligo-anuric state as important reason for hyperphosphatemia among patients with advanced CKD in the discussion of our manuscript a as suggested. The following text has been added.
“In addition to oligo-anuric state, patients with advanced CKD/ESKD have a significant reduction in klotho and progressively lose the ability to prevent phosphate retention, resulting in hyperphosphatemia, vascular calcification, and cardiovascular disease (83, 84).”
Comment #2
2.As regards the reason post-KTx hypophosphatemia, the study showed that post-KTx Klotho levels are lower than controls, while FGF23 similar to the control. Thus, the given explanation only partially can be the reason for post-KTx hypophosphatemia. And certainly due to ischemia-reperfusion injury, the quality of the grafts is not more excellent than of a native normal kidney. Initially, post-KTx hypophosphatemia can be partly attributed to the previously accumulated FGF23, but there are also other more persistent reasons. Residual hyperparathyroidism, which is not uncommon, is another explanation. Cyclosporine may also induce hypophosphatemia since it inhibits the proximal tubular brush border membrane Na/Pi cotransport activity in rats. A study detected faster recovery of impaired phosphate reabsorption by the proximal tubule in patients treated with tacrolimus compared to cyclosporine. These points were not evaluated and should be discussed reasonably instead of indirectly, through hypophosphatemia, attribute the better outcome to partial Klotho restoration after KTx.
Response: The reviewer raises very important point. We appreciate and agree with this point. We have additionally included this point as reviewer’s suggestion and also included references of cyclosporine induced hypophosphatemia in rats and also in clinical study as reviewer’s suggestion. The following text has been added.
“Following successful KTx, patients regain functions of klotho via FGF23-Klotho signaling, and with the previously accumulated FGF23, residual hyperparathyroidism, the use of calcineurin inhibitors (especially cyclosporine) (93-95), post-KTx hypophosphatemia can commonly occur up to 86% (85, 96, 97).”
We greatly appreciated the editor and reviewer’s time and comments to improve our manuscript.

Reviewer 2 Report
In this paper Charat Thongprayoon et al, performing a systematic review and meta-analysis, concluded that there was a significant reduction in serum klotho levels after living kidney donation and an increase in serum klotho levels after KTx.
Although the paper is well written, I believe that the low number of studies included (all observational and extremely heterogeneous) and the absence of an accurate control of confounding factors (the results are only partially adjusted for clinical covariates), make this study, at this stage, not strong enough to provide definitive conclusions and to be publishable on JCM journal. Too many variables may influence klotho levels in the post-transplant period.
However, I encourage authors to use these data as starting point for a future prospective study to assess the impact of changes in klotho on clinical outcomes post-KTx.
Author Response
Response to Reviewer#2
In this paper Charat Thongprayoon et al, performing a systematic review and meta-analysis, concluded that there was a significant reduction in serum klotho levels after living kidney donation and an increase in serum klotho levels after KTx.
Although the paper is well written, I believe that the low number of studies included (all observational and extremely heterogeneous) and the absence of an accurate control of confounding factors (the results are only partially adjusted for clinical covariates), make this study, at this stage, not strong enough to provide definitive conclusions and to be publishable on JCM journal. Too many variables may influence klotho levels in the post-transplant period.
However, I encourage authors to use these data as starting point for a future prospective study to assess the impact of changes in klotho on clinical outcomes post-KTx.
Response: We thank you for reviewing our manuscript and for your critical evaluation. The reviewer raises very important point. As guest editors of JCM special issue, we appreciate the reviewer’s important comments to improve our manuscript for this special issue and we are hoping to have opportunity to request input from the reviewer again on a potential future prospective study on this topic. We agree with the expert reviewer. Although there is no significant heterogeneity in analysis assessing serum klotho after living kidney donation, the reviewer raises very important point regarding the heterogeneity. We thus have additionally included discussion on heterogeneity between the included studies evaluating changes in serum klotho levels among KTx recipients in the limitations of our study as suggested.
- Also, we agree with the reviewer and have additionally emphasized in the discussion of fulltext and in the abstract that future prospective studies are needed to assess the impact of changes in klotho on clinical outcomes in KTx recipients and living kidney donors, as suggested. We are hoping that the findings from our study will further motivate the research to advance the knowledge in the field of Klotho in kidney transplant recipients and live donors.
- We agree with the reviewer regarding factors that may influence klotho levels in the post-transplant period. We reviewed all included studies again and performed additionally analysis to provide more details on changes of PTH, Ca, and Phos levels after kidney transplant recipients, in addition to serum klotho and FGF-23 levels.
-We also make the data for this meta-analysis publicly available, through the Open Science Framework ((URL: https://osf.io/kx9we/).) to promote the reproducibility of this meta-analysis.
The following text has been added as suggested.
“Second, many variables may influence klotho levels in the post-transplant period that may contribute to the heterogeneity between the included studies evaluating changes in serum klotho levels among KTx recipients. Data on medications that may affect endogenous klotho expression in the kidney and soluble levels such as angiotensin II inhibitors and hydroxymethylglutaryl-CoA (HMG-CoA) reductase inhibitors (17, 21, 119, 120) as well as data on immunosuppression were limited in included studies. Lifestyle, diet, psychological stress, and activities such as exercises may also affect serum klotho levels (121-124). Thus, future prospective studies are needed to assess the impact of changes in klotho on clinical outcomes in KTx recipients and living kidney donors.”
“There were significant reductions in serum PTH and phosphate levels with MDs of – 134.65 pg/mL (95%CI – 176.09 to -93.21 pg/mL, I2 = 0%) and – 2.81 mg/dL (95%CI – 3.46 to -2.16 mg/dL, I2 = 97%), respectively. There was no significant change in serum calcium levels with MD of 0.37 mg/dL (95%CI, - 0.05 to 0.79 mg/dL, I2 = 83%).”
“The data for this meta-analysis are publicly available through the Open Science Framework (URL: https://osf.io/kx9we/).”
We greatly appreciated the editor and reviewer’s time and comments to improve our manuscript.

Round 2
Reviewer 1 Report
The authors addressed my comments adequately.
Reviewer 2 Report
Although I still think it is too early to carry out a meta-analysis on this topic (too few studies and too many post-tx variables to be considered), authors have clearly reported major limitations of this study in the manuscript text. Therefore, after revision, the paper is conceptually improved and should be considered for publication in JCM.